# Hybridized Zoonotic *Schistosoma* Infections Result in Hybridized Morbidity Profiles: A Clinical Morbidity Study amongst Co-Infected Human Populations of Senegal

**DOI:** 10.3390/microorganisms9081776

**Published:** 2021-08-20

**Authors:** Cheikh B. Fall, Sébastien Lambert, Elsa Léger, Lucy Yasenev, Amadou Djirmay Garba, Samba D. Diop, Anna Borlase, Stefano Catalano, Babacar Faye, Martin Walker, Mariama Sene, Joanne P. Webster

**Affiliations:** 1Service de Parasitologie—Mycologie, Faculté de Médecine, Pharmacie et Odontologie, Université Cheikh Anta Diop, Dakar BP 5005, Senegal; chek2810@yahoo.fr (C.B.F.); bfaye67@yahoo.fr (B.F.); 2Centre for Emerging, Endemic and Exotic Diseases, Department of Pathobiology and Population Sciences, Royal Veterinary College, University of London, Herts AL9 7TA, UK; eleger@rvc.ac.uk (E.L.); lyasenev@hotmail.com (L.Y.); anna.borlase@bdi.ox.ac.uk (A.B.); ste.cata85@gmail.com (S.C.); mwalker@rvc.ac.uk (M.W.); 3London Centre for Neglected Tropical Disease Research (LCNTDR), Faculty of Medicine (St Mary’s Hospital Campus), Imperial College, London W2 1PG, UK; 4Réseau International Schistosomoses, Environnement, Aménagement et Lutte (RISEAL-Niger), Niamey BP 13724, Niger; garbadjirmaya@who.int; 5Institut Supérieur de Formation Agricole et Rurale, Université de Thiès, Bambey BP 54, Senegal; sambadeguene.diop@gmail.com; 6Unité de Formation et de Recherche des Sciences Agronomiques, d’Aquaculture et de Technologies Alimentaires, Université Gaston Berger, Saint-Louis BP 234, Senegal; mariama-sene.wade@ugb.edu.sn

**Keywords:** hybridization, schistosomiasis, morbidity, ultrasonography, disease control, one health

## Abstract

Hybridization of infectious agents is a major emerging public and veterinary health concern at the interface of evolution, epidemiology, and control. Whilst evidence of the extent of hybridization amongst parasites is increasing, their impact on morbidity remains largely unknown. This may be predicted to be particularly pertinent where parasites of animals with contrasting pathogenicity viably hybridize with human parasites. Recent research has revealed that viable zoonotic hybrids between human urogenital *Schistosoma haematobium* with intestinal *Schistosoma* species of livestock, notably *Schistosoma bovis,* can be highly prevalent across Africa and beyond. Examining human populations in Senegal, we found increased hepatic but decreased urogenital morbidity, and reduced improvement following treatment with praziquantel, in those infected with zoonotic hybrids compared to non-hybrids. Our results have implications for effective monitoring and evaluation of control programmes, and demonstrate for the first time the potential impact of parasite hybridizations on host morbidity.

## 1. Introduction

Hybridization amongst parasitic agents, particularly concerning those with zoonotic potential, is a major emerging public and veterinary health concern at the interface of evolution, epidemiology, ecology, and control. Co-infections, where individual hosts are infected by more than one infectious agent at the same time, are the norm within humans and animals [1]. Increasing levels of anthropogenic changes are shifting the opportunities for encountering new infections of both human and animal origin [2], and thereby also the occurrence of co-infections with multiple agent species and strains [3]. Co-infection can have a significant impact on the pathogens involved, often as a result of synergistic or antagonistic interactions, where changes in establishment, growth, maturation, reproductive success, and/or drug efficacy have all been documented. Furthermore, co-infections between parasites can allow for heterospecific (between-species or between-lineage) mate pairings, resulting in either infertility or parthenogenesis (asexual reproduction where eggs occur without fertilization), introgression (the introduction of alleles of one species into the gene pool of another through repeated backcrossing of an inter-specific hybrid with one of its parent species), or whole genome admixture [3]. Molecular developments have revealed an increasing number of fertile hybridization and introgression events across humans, animals, and also eukaryotic parasites [3,4,5,6,7]. Parasites are especially valuable models for studying speciation and introgression processes, due to their intimate association with their host organisms [6]. However, the potential impact of inter-specific parasite hybridizations on host morbidity remains almost entirely unknown. This may be particularly pertinent when one considers the potential for parasites of animals to viably hybridize and introgress with parasites of humans, and where such parasites may be responsible for highly contrasting morbidity profiles in their original single species form or host.

One group of infectious agents where opportunities for, and subsequent evidence of, hybridization between parasites of humans with those from animals is rapidly emerging are those of the neglected tropical diseases (NTDs)—highly debilitating diseases infecting more than a fifth of the world’s human population, and their livestock, with devastating consequences. One such major NTD is schistosomiasis, the second most important parasitic disease of humans, after malaria, in terms of socio-economic impact. More than 240 million people are currently infected [8], with an estimated minimum of 200,000 deaths annually within sub-Saharan Africa (SSA) [9,10]. Environmental and anthropogenic changes (e.g., dam constructions), and the movement of parasitized people and animals, have all served to facilitate the introduction of new schistosome species to new areas [11,12] and/or different species co-infecting the same host [13]. For instance, the human intestinal schistosomiasis species *Schistosoma mansoni* (*Sm*) and the human urogenital schistosomiasis species *Schistosoma haematobium* (*Sh*) are both prevalent across much of SSA, representing opportunities for co-infections [14,15,16,17,18], and are also reported amongst migrants globally, such as across China [19] and Europe [20,21,22]. Furthermore, recent developments in molecular techniques have revealed the—often both extensive and persistent—existence of viable zoonotic hybridization and introgression between the human urogenital schistosome species *Sh* and closely-related intestinal schistosome species of livestock, such as *Schistosoma bovis* (*Sb*)*, Schistosoma curassoni* (*Sc*), and/or *Schistosoma mattheei* (*Smt*) across parts of SSA [23,24,25,26,27,28], and even, at least as regards *Sh-Sb* hybrids, within parts of Europe (Corsica) [12,29].

Whilst the impact of these hybridized parasites on morbidity profiles is unknown, clinical manifestations of schistosomiasis are primarily associated with the species-specific oviposition site. After entering the human host, schistosomes mature to adulthood in the liver, with the female residing within the gynaecophoric canal of the larger male. Schistosome males then carry the pair to the mesenteric plexus for intestinal species such as *Sm*, and to the veins of the pelvis for the urogenital *Sh*. Females shed eggs into the blood, some being expelled to the external environment within either stool (*Sm*) or urine (*Sh*). However, a large proportion of these spined eggs remain trapped within the host’s tissues and induce granulomatous inflammatory reactions leading to local and systemic pathological effects. Intestinal schistosomiasis can cause severe hepatosplenomegaly and periportal fibrosis with portal hypertension. Hematuria and urogenital inflammation, including bladder cancer and lesions in the reproductive system are characteristics of urogenital schistosomiasis [30,31]. In children, schistosomiasis, in part through continued inflammation, has been reported to impede normal growth, iron metabolism, physical fitness, and cognitive function, with consequent disabling systemic morbidities including anemia, malnutrition, and impaired development [31].

Whilst disease control programmes consider species-specific morbidity in isolation, research across SSA has found that *Sm* with *Sh* co-infections result in lower hepato-splenic morbidity compared to single *Sm* infections but increased urogenital morbidity as compared to single *Sh* infections [14,16,17]. This is believed to occur because dominant *Sh* males divert *Sm* females from the portal vein to the vesical plexus, resulting in less eggs in liver tissues but more eggs in bladder tissues [14,16,17,32]. Similar mechanisms may be predicted to impact morbidity patterns in zoonotic hybrid infections (i.e., those such as *Sh-Sb*, *Sh-Sc* or *Sh-Smt*), given the pairing of a human urogenital schistosome species with an animal intestinal species. Furthermore, hybrid vigor may be predicted to occur, resulting in increased egg production relative to single-species infections, as has been hinted by some animal studies [33]. Hybridization could thus represent a significant, but previously ignored, issue for the monitoring and evaluation of morbidity during mass drug administration programmes and a significant challenge to the WHO targets towards elimination of schistosomiasis as a public health problem by 2030 [34]. More generally, inter-specific hybridization between urogenital pathogens of humans with those of intestinal pathogens of animals may be predicted to have broader implications and applications for predicting complex morbidity profiles amongst both ongoing endemic and future epidemic diseases.

In this study, through examining children and adults from highly disease endemic mixed foci regions of Senegal, West Africa, we predicted that *Sh-Sb* hybrid infections would be associated with differential and/or exacerbated host morbidity profiles as compared with single-species infections (whilst also controlling for *Sh* with *Sm* coinfections). We also evaluated the relationship between the impact of praziquantel treatment on morbidity and the infecting species’ combination(s), where we predicted that exacerbated morbidity for mixed or hybrid infections would result in less morbidity resolution following treatment as compared to single or non-hybridized species infections.

## 2. Materials and Methods

### 2.1. Study Sites and Populations

The study was conducted in two areas of northern Senegal: Richard Toll (RT), where schistosomiasis transmission is perennial and *Sm*, *Sh*, and zoonotic *Sh-Sb* hybrids are co-endemic [26,27]; and Barkedji (BK), a focus of human *Sh* urogenital schistosomiasis with less frequent cases of zoonotic *Sh-Sb* and *Sh-Sc* hybrid infections and where transmission is seasonal [23,27]. Full details on the study sites and population infection prevalence and intensities have been published in Léger et al. (2020) [23].

We performed two surveys (Figure 1) in May–August 2016 (subsequently referred to as “2016”) and October 2017–January 2018 (“2017”). Children aged 5–17 years (*n* = 1319) were randomly selected from school registers with the sample size for each school proportional to the number of children registered. Adults aged 18–78 years (*n* = 300) were self-selecting volunteers. The surveys were carried out approximately six months after the latest mass drug administration activities carried out by the Senegalese national schistosomiasis control programme (December 2015 for both RT and BK, and December 2016–January 2017 for RT only).

Parasitological and clinical examinations were undertaken in 2016 and 2017, whereas ultrasonography was performed only in 2017. All individuals diagnosed with schistosomiasis were treated with 40 mg/kg of praziquantel. A follow-up was conducted approximately one month after treatment (before any reinfecting schistosomes reached egg-laying maturity), with the same parasitological, clinical and ultrasonographical examinations as pre-treatment (*n* = 414).

It was not possible to reconstruct longitudinal cohorts between 2016 and 2017 because of anonymization requirements at each sampling time (please see Statistical Analyzes and Ethics statement). For this reason, re-infection of individuals in between 2016 and 2017 treatment events could not be assessed but was highly likely in this endemic setting.

### 2.2. Parasitological and Molecular Procedures

Details of the parasitological and molecular procedures are described in detail in Léger et al. (2020) [23]. Briefly, one urine and one stool sample were collected in the morning from each individual. Two 10 mL urine filtrations and two Kato–Katz slides were performed on each sample to detect and count schistosome eggs. Urine (filtration) and fecal egg counts (Kato–Katz) were standardized to eggs per 10 mL of urine (ep 10 mL) or eggs per gram of feces (epg), respectively, and were defined as urogenital or intestinal infections, respectively. DNA was extracted from individual miracidia (collected by miracidia hatching techniques from positive urine and stool samples) stored on Whatman Indicating FTA Classic Cards (GE Healthcare Life Sciences, Buckinghamshire, UK). DNA extracts were characterized by amplification of a partial fragment of the mitochondrial cytochrome *c* oxidase subunit 1 (*cox1*) and the complete nuclear ribosomal DNA internal transcribed spacer (ITS). All individual miracidia were classified as hybrids using both ITS and *cox1* genotyping simultaneously (samples with only one of the two mitochondrial or nuclear ribosomal measures amplifying were discarded as inconclusive and/or to prevent potential bias regarding hybridizing directionality). Although the combined *cox1*-ITS approach lacks the precision to reveal the full history of hybrid populations and may miss highly backcrossed or introgressed hybrid lineages relative to genomic approaches [35], it has been repeatedly used to successfully identify various stages of hybridization in natural populations and has the advantage of allowing larger scale sampling [23,25,26,27,36]. Furthermore, as early generation hybrids might be more virulent and causing more severe or atypical morbidity [3,37,38], this method is particularly relevant here in the context of our study. The number of miracidia successfully sampled and typed per individual host ranged from one to fifteen (minimum eight miracidia typed per host, or all if less than eight were available) [23]. Although it is possible that individuals with hybrids were missed given the sample sizes, the strategy was to increase the number of hosts with genotyped miracidia, rather than the number of genotyped miracidia per host, to better reflect the genetic diversity of the parasite populations [23,39]. Overall, the frequency of miracidia classified as hybrids was 23.2% (*n* = 2538 miracidia with both ITS and *cox1* genotyped).

### 2.3. Clinical Procedures

Urine strips (Hemastix, Siemens Healthcare Diagnostics, Surrey, UK) were used to determine the presence of blood in the urine (hematuria). The results were expressed ordinally as −, +, ++ or +++. Hemoglobin levels in the blood were assessed using a HemoCue device (Agelholm, Sweden). Anemia was defined as none, mild, moderate or severe using World Health Organization (WHO) hemoglobin thresholds (Appendix A Appendix A) [40].

Ultrasound examinations of urogenital and intestinal schistosomiasis were performed according to the standardized WHO Niamey protocol [41], by a single experienced sonographer using a sector probe (Convex 3C6C). For urogenital schistosomiasis, standard examinations of the bladder, the ureters and the kidneys were carried out to detect lesions with: (i) a cross section of the bladder to highlight the different types of lesions (irregularity, thickening, mass, pseudo-polyp, calcification); (ii) a view of the distal part of the ureters; and (iii) a left and right lateral section centered on the kidneys and the proximal part of the ureters. Each person drank water or juice half an hour before the exam for the filling of the bladder, which is essential for examining its shape and irregularities in the wall. The urinary bladder intermediate score (UBIS) was calculated for specific lesions of the bladder which are 0–1: unlikely, 2: likely and ≥3: very likely due to schistosomiasis [41]. The upper urinary tract intermediate score (UTIS) was calculated for lesions of the ureters and kidneys, associated with severe morbidity but not specific to schistosomiasis. The global score (GS) was obtained by adding the two intermediate scores for each individual [41]. UTIS and GS were categorized as negative (=0) or positive (≥1).

Investigations of lesions due to intestinal schistosomiasis were performed on individuals that had fasted for at least four hours before the ultrasound examination [41]. The liver was compared to image patterns A–F to assess for periportal fibrosis (Appendix A Appendix A) [41,42]. Measurement of the size of the left and right liver lobes, of the internal diameter of the portal vein and of the spleen, were categorized as normal, moderately abnormal or severely abnormal, depending on the participant’s height [41].

### 2.4. Statistical Analyses

We used cumulative link logistic regression models (Appendix A) to analyze the impact of the presence and intensities of current urogenital and intestinal infections and their interaction on each pre-treatment morbidity indicator. This analysis was conducted for hematuria and anemia for children (*n* = 602) and adults (*n* = 129) in 2016, and for hematuria, anemia and lesions observed using ultrasound for children (*n* = 724) and adults (*n* = 175) in 2017 (Appendix A Appendix A). Following WHO infection intensity definitions [43], we assigned 0 for not infected (0 ep 10 mL), 1 for light (<50 ep 10 mL) and 2 for heavy (≥50 ep 10 mL) urogenital infections, and 0 for not infected (0 epg), 1 for light (1–99 epg), 2 for moderate (100–399 epg) and 3 for heavy (≥400 epg) intestinal infections. We assumed linear effects of infection intensity classes [16], which were tested using the Akaike Information Criterion (AIC): models assuming numerical values of infection intensity classes consistently performed better in terms of AIC compared to models fitting separate effects for each combination of urogenital and intestinal intensity category.

We used logistic regression models to analyze the association between decreases in morbidity indicators following praziquantel treatment, and the presence and pre-treatment intensities of urogenital and intestinal infections. Only individuals who already had positive pre-treatment morbidity indicators (i.e., individuals with anemia, hematuria, or lesions observed using ultrasound) and successful follow up one month after treatment were included. The number of individuals successfully followed up was too small to perform this logistic regression for children in 2016 and for adults in both surveys. Therefore, this analysis was conducted only for children in 2017 (*n* = 334—Appendix A Appendix A), and interactions between urogenital and intestinal infection intensities were not included because of insufficient data.

We repeated the analyses using the subset of data from those individuals contributing genotyped miracidia to determine whether current species–species combinations were associated with each morbidity indicator. Because of small sample sizes available from adults, these analyses were conducted only for children (*n* = 203 children in 2016 and 223 children in 2017 for pre-treatment cumulative link logistic regression models—Appendix A Appendix A; *n* = 179 children in 2017 for post-treatment logistic regression models—Appendix A Appendix A).

In all analyses, the effects were adjusted for age, sex and study site, and school was defined as a random effect for children to account for any clustering of morbidity indicators. Analyses for hematuria and anemia were conducted separately for each sampling year because it was not possible to reconstruct longitudinal cohorts due to ethical requirements for anonymization at each sampling time.

All statistical analyses were performed in R version 3.6.2 (R Core Team, Vienna, Austria), using the ‘ordinal’ package to fit the cumulative link logistic regression models [44,45,46].

### 2.5. Ethics Statement

Ethical approval was provided by: (i) the Imperial College (London, UK) application 03.36; (ii) the Royal Veterinary College (London, UK) application URN20151327; and (iii) the Comité National d’Ethique pour la Recherche en Santé (Dakar, Senegal) application SEN15/68. Written informed consent was obtained from all adult participants, children’s parents, or guardian, with additional verbal consent provided by the children. All data were fully anonymized prior to analysis.

## 3. Results

### 3.1. Prevalence of Infection and Morbidity

Full details on the epidemiological findings in terms of *Schistosoma* hybrid/species prevalence, intensities and distributions have been published in Léger et al. (2020) [23]. Briefly, the prevalence of urogenital schistosomiasis was higher in RT compared to BK, with both *Sh* and *Sh-Sb* hybrids identified [23] (Appendix A Appendix A). Higher proportions of *Sh-Sb* hybrids were found in RT than in BK [23] (Appendix A Appendix A). Intestinal schistosomiasis was found only in children from RT at low prevalence, and only *Sm* was identified except for three children that presented eggs genotyped as *Sh* in their stool [23] (Appendix A Appendix A). No *Sh-Sb* hybrids were retrieved from stool, and no ectopic *Sm* eggs were found in urine. Almost all children from RT who were positive for *Sm* were co-infected with *Sh* or *Sh-Sb* urogenital schistosomiasis: 89% (39/44) in 2016 and 92% (34/37) in 2017.

Morbidity indicators in children were consistently higher in RT compared to BK, except for anemia in 2016 (Table 1). Lesions of the urogenital tract were more frequently found in the bladder than in the upper tract (Table 1), the main lesions being irregularities, wall thickening and masses of the bladder wall, unilateral and bilateral dilatations of the ureters, and unilateral dilatations of the kidneys (Appendix A Appendix A and Appendix A). Anormal size of the right liver lobe, dilatation of the portal vein and splenomegaly were rare or absent (Table 1). All liver images assessing for periportal fibrosis were classified as normal (image pattern A in the Niamey protocol; Appendix A Appendix A).

### 3.2. Impact of Single and Mixed Urogenital and Intestinal Infection on Morbidity

All morbidity indicators relating to the urogenital tract had higher odds of being positive and severe in children with single urogenital infection compared to uninfected children, and in children with heavy compared to light infection intensity (Table 2 and Table 3), but this was not the case for anemia or hepatomegaly (Appendix A Appendix A). Similar results were found in adults (Appendix A Appendix A). No differences were found between the different lesions of the urinary bladder wall (irregularities, thickening and masses; Appendix A Appendix A). Lesions of the ureters were associated with urogenital infection presence and intensity, but not lesions of the kidneys (Appendix A Appendix A).

Presence and intensity of single *Sm* intestinal infections were not statistically significantly associated with any of the morbidity indicators (Table 3). However, only 37 children were infected with *Sm* in 2017, among which only three were singly infected.

Estimates of the interaction between the intensity of urogenital and intestinal schistosomiasis indicated that *Sh* with *Sm* co-infection was not associated with the morbidity indicators, although there was a marginal (but not statistically significant) increase in UBIS and bladder wall calcification (odds ratios: 2.75 (0.79–9.54), *p* = 0.11 and 3.25 (0.85–12.4), *p* = 0.084, respectively).

Independent of infection status, greater odds of positive and severe morbidity indicators were estimated for children in RT compared to BK, except for hematuria in 2016, and in boys compared to girls, except for hematuria and hepatomegaly (Appendix A Appendix A). Differences between sites and sex were not found in adults, except that hematuria was more prevalent and severe in women than in men (Appendix A Appendix A).

### 3.3. Impact of Infection by Sh-Sb Hybrids on Morbidity

Overall, there was no difference between *Sh-Sb* hybrid and non-hybrid *Sh* infections on the odds of positive urogenital morbidity indicators (Figure 2). There was a non-significant decrease in the odds of positive UTIS in children with *Sh-Sb* hybrids (Figure 2). This effect was significant for lesions of the ureters, but not for lesions of the kidneys (Figure 2). Notably, children with *Sh-Sb* hybrid infections had significantly higher odds of hepatomegaly than individuals with non-hybrid *Sh* infections (Figure 2).

### 3.4. Impact of Treatment on Infection and Morbidity

The cure rates (percentage of participants infection-negative after treatment who were infection-positive before treatment [47]) and egg reduction rates (percentage reduction in intensity of infection after treatment [47]) of urogenital schistosomiasis following praziquantel treatment were high in both study sites in both surveys (Table 4).

One month post-treatment, morbidity indicators decreased or became negative for the majority of children and adults who had positive morbidity indicators pre-treatment (without considering schistosome species or combination of species) (Table 5). These proportions were not significantly different between sites. Changes in morbidity following treatment were dependent on the presence and intensity of urogenital infection before treatment only for GS and UBIS, which decreased more frequently in children with urogenital infection compared to uninfected children, and in children with heavy compared to light infection intensity (Table 6). Most children whose morbidity scores did not decrease had low values of 1 or 2: 90% (52/58) for GS and 98% (64/65) for UBIS. No significant effects of age, sex or study site were detected.

Changes in morbidity were also independent of the presence or absence of *Sh-Sb* hybrids, except for anemia, which tended to decrease more frequently in children with *Sh-Sb* hybrids than in children with non-hybrid *Sh* infection, and for GS and UBIS, which decreased less frequently in children with *Sh-Sb* hybrids than in children with non-hybrid *Sh* (Figure 3). Most children whose GS or UBIS scores did not decrease had low values of 1 or 2: 91% (29/32) for GS and 100% (36/36) for UBIS.

## 4. Discussion

As the distribution of human, domestic animal and wildlife parasites is modified by anthropogenic and environmental changes, the frequency of co-infection and hybridization events is likely to increase [3]. Understanding their impact on clinical manifestations and treatment efficacy is therefore critical to inform public health decision-making, as well as help predict the consequences of inter-specific parasite hybridization for pathogenesis in general.

Here, we focused on the clinical manifestations induced as a consequence of hybridization of the human urogenital *Sh* with the livestock intestinal schistosome species *Sb*, as well as interactions between the two major human schistosome species, *Sh* and *Sm*, in two areas of northern Senegal. We uniquely demonstrated, in line with our predictions, increased hepatic morbidity (hepatomegaly) and decreased urogenital morbidity (ureteral lesions) in *Sh*-*Sb* hybrid infections compared to non-hybrid *Sh* infections, whilst controlling for *Sm* co-infections. Furthermore, whilst treatment with praziquantel was, in general, effective in rapidly reducing morbidity, regardless of infection status, bladder lesions decreased less frequently in response to praziquantel in children with *Sh-Sb* hybrids relative to *Sh* single species infections. Although the vast majority of children that did not respond to treatment had low scores of 1 or 2, possibly reflecting small lesions not due to schistosomiasis [41], this could suggest that bladder lesions induced by *Sh-Sb* hybrids could resolve after a longer period than the one month which was evaluated here, or may require further praziquantel treatments. Therefore, further research is needed to assess the success of morbidity control and treatment efficacy in *Sh-Sb* hybrid infections.

Although morbidity may depend on cumulative exposure and treatment history, we observed strong positive relationships between current urogenital infection intensities and bladder and upper urinary tract lesions as well as hematuria, in accordance with previous studies [16,48,49,50,51,52]. We also observed a tendency of increased bladder morbidity in *Sh* with *Sm* co-infections compared to single *Sh* infections, in accordance with Malian and Kenyan studies [14,16]. In contrast, a previous study in northern Senegal suggested a protective effect of *Sm* co-infections on bladder morbidity, when *Sm* eggs are eliminated exclusively via urine (ectopic eggs) [48]. We did not find any evidence of such a protective effect, as no *Sm* ectopic eggs were found in our study (*Sm* eggs were eliminated only in feces here).

Boys tended to have more bladder and upper tract lesions compared to girls [49,52], with urogenital morbidity indicators more frequent and severe among children in the site where transmission is perennial (RT) compared to seasonal transmission site (BK) [49]. In contrast, no differences between sexes and sites were found in adults, except for hematuria being more frequent in women than in men, potentially also an artifact of menstruation.

Notably, we also observed different patterns for lesions of the upper urinary tract and lesions of the bladder. Indeed, the positive relationship between urogenital infection intensities and lesions was not as strong when considering the UTIS, and was not evidenced at all when considering kidneys alone. As upper tract lesions are associated with severe morbidity [41], we thereby further emphasize here a need to distinguish between the two intermediate scores and their individual components when assessing schistosomiasis morbidity, rather than using the synthetic global score. This distinction between lower and upper urinary tract has generally been overlooked in past studies, which have tended to consider only positive GS or only the UBIS [16,48,49,51]. By contrast, we did not find any difference between the main bladder lesions and the prevalence of bladder lesions, consistent with previous studies [48,49,52].

Our observation that *Sh*-*Sb* hybrids are associated with higher odds of hepatomegaly relies on a relatively small subset of data (*n* = 42). Although we accounted for urogenital and intestinal infections presence and intensity, age, sex, and study site, possible sampling biases for other confounders prevalent in Senegal could not be excluded, such as pulmonary hypertension [53], hepatitis B [54], sickle-cell disease [55] or malnutrition [56]. Although malaria is a major cause of hepatomegaly, the prevalence of malaria is very low in the study area and therefore is not a likely confounder, at least amongst the children. It is also possible that some individuals with hybrids, in particular with introgressed or backcrossed ones, were missed either during miracidia hatching or genotyping.

In *Sh* and *Sm* co-infections, differential morbidity profiles have been proposed to be the consequence of dominant *Sh* males diverting *Sm* females from the portal vein to the vesical plexus, resulting in fewer eggs in liver tissues but more eggs in the bladder tissues [14,16,32]. A similar process may be predicted in zoonotic hybrid infections if dominant *Sh* males shift location of *Sh*-*Sb* females. However, there is evidence that hybridization between *Sh* and *Sb* can be both bidirectional [25,27] and that the majority of current hybrids, at least in some regions, may be the result of ancient introgression [57]. Therefore, the morbidity profiles for *Sh*-*Sb* hybrids are less predictable, as *Sh*-*Sb* males, for example, could shift *Sh* females resulting this time in more eggs in the liver and fewer eggs in urogenital tissues. Whilst the latter mechanism could explain the hybrid morbidity profiles we observed, with increased hepatic morbidity and decreased urogenital morbidity, it is unclear how to explain that *Sh-Sb* eggs were only shed in urine here [23] (although *Sh-Sb* eggs have been previously detected in stool from Senegalese children [25]). As all schistosome species mature and form male–female pairs in the liver, one possible explanation may thus be that *Sh-Sb* eggs are more likely to become trapped in the liver and, therefore, cause more morbidity associated with intestinal schistosomiasis, albeit with no or few eggs expelled in feces. These results emphasize the need to measure both hepatic and urogenital morbidity indicators during mass drug administration monitoring and evaluation wherever co-infections and/or notably zoonotic hybrid species are suspected. Moreover, whereas current practice is simply to assume eggs in urine represent *Sh* whilst those from stool reflect *Sm,* we strongly advise to use, where logistically feasible, molecular identification to identify what morbidity profiles are attributable to which species, combination of species and/or hybrids.

## 5. Conclusions

To conclude, our results suggest differential morbidity profiles in hybrid infections relative to single species infections. Hybridization thus represents a significant issue for the monitoring and evaluation of morbidity within mass drug administration programmes. More broadly, our findings highlight the importance of integrating parasitology, molecular and evolutionary biology, and medicine to elucidate and predict the consequences of parasite hybridization for clinical infectious disease management and prevention in general.

## Figures and Tables

**Figure 1 microorganisms-09-01776-f001:**
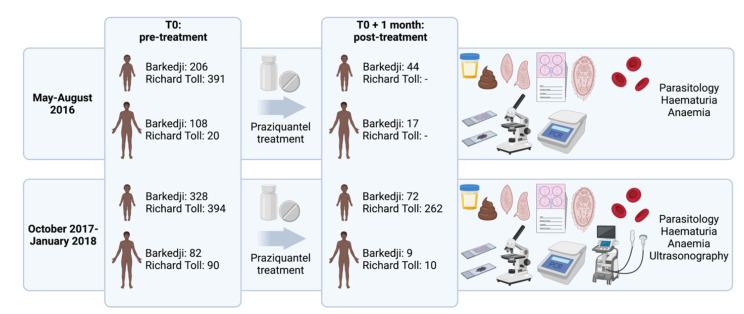
Sampling design of the study conducted in two areas of northern Senegal: Barkedji and Richard Toll. Only parasitological (urine filtration, Kato–Katz, miracidial hatching) and clinical examinations (hematuria and anemia) were performed during the May–August 2016 survey, whereas parasitological, clinical and ultrasonographical examinations were performed during the October 2017–January 2018 survey. School-aged children (*n* = 1319) were randomly selected, while adults (*n* = 300) were self-selecting volunteers. All individuals diagnosed with schistosomiasis were treated with 40 mg/kg of praziquantel. For both surveys, a follow-up was conducted approximately one month after treatment, with the same examinations as pre-treatment (i.e., parasitological and clinical examinations in 2016, *n* = 61; parasitological, clinical and ultrasonographical examinations in 2017, *n* = 353). Numbers indicate individuals that received at least one of the parasitological or clinical examinations. Created with BioRender.com (3 August 2021).

**Figure 2 microorganisms-09-01776-f002:**
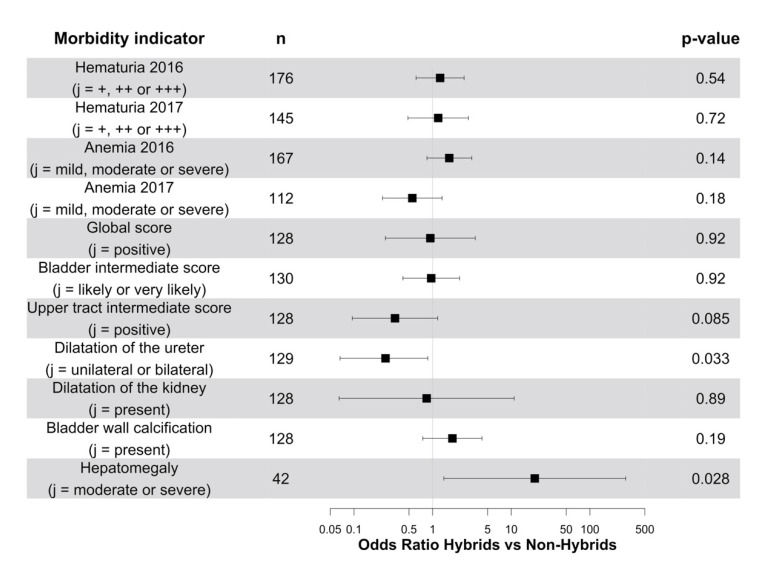
Odds ratios (points) and 95% confidence intervals (horizontal lines) of various morbidity indicators being in category *j* or above among children in Senegal in 2016 and 2017, depending on the detection of *S. haematobium-S. bovis* hybrids and adjusted for urogenital and intestinal schistosomiasis infection intensities, age, sex, and study site. For each morbidity indicator, only individuals with genotyped miracidia and complete data on infection intensity, age, sex, and study site were included. Urine strips were used to determine the presence of blood in the urine (hematuria), and the results were expressed as −, +, ++ or +++, indicative of increasing intensity.

**Figure 3 microorganisms-09-01776-f003:**
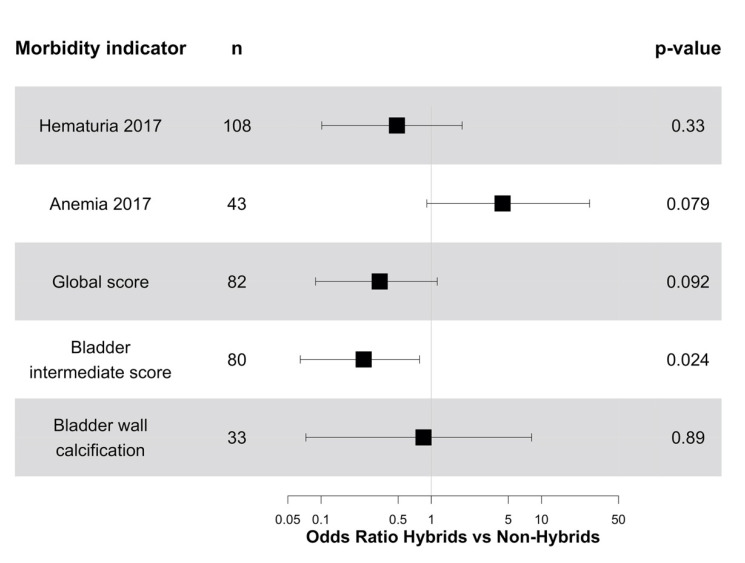
Odds ratios (points) and 95% confidence intervals (horizontal lines) of decreased levels of morbidity in response to treatment with praziquantel among children in Senegal in 2017, depending on the detection of *S. haematobium-S. bovis* hybrids pre-treatment and adjusted for intestinal and urogenital infection intensities before treatment, age, sex, and study site.

**Table 1 microorganisms-09-01776-t001:** Prevalence of morbidity indicators measured in children and adults in two study sites of Senegal in 2016 and 2017 (number of individuals examined indicated in parentheses).

Year	Morbidity Indicator	Children		Adults	
		Barkedji	Richard Toll	Barkedji	Richard Toll
2016	Hematuria	34% (*n* = 203)	62% (*n* = 376)	40% (*n* = 105)	75% (*n* = 20)
	Anemia	43% (*n* = 148)	51% (*n* = 382)	36% (*n* = 45)	-
	Total examined ^1^	*N* = 203	*N* = 391	*N* = 108	*N* = 20
2017	Hematuria	28% (*n* = 326)	73% (*n* = 368)	40% (*n* = 80)	39% (*n* = 82)
	Anemia	-	54% (*n* = 374)	-	74% (*n* = 43)
	Global score ^2^	25% (*n* = 297)	65% (*n* = 335)	32% (*n* = 19)	41% (*n* = 41)
	Bladder intermediate score	21% (*n* = 303)	48% (*n* = 335)	16% (*n* = 19)	15% (*n* = 41)
	Upper tract intermediate score	5% (*n* = 297)	15% (*n* = 335)	0% (*n* = 19)	2% (*n* = 41)
	Bladder wall calcification	6% (*n* = 297)	27% (*n* = 335)	0% (*n* = 19)	20% (*n* = 41)
	Hepatomegaly	69% (*n* = 109)	89% (*n* = 99)	0% (*n* = 2)	100% (*n* = 12)
	Abnormal size of the right liver lobe	0.7% (*n* = 144)	2.3% (*n* = 222)	50% (*n* = 2)	0% (*n* = 44)
	Dilatation of the portal vein	0% (*n* = 147)	0% (*n* = 220)	0% (*n* = 2)	0% (*n* = 46)
	Splenomegaly	0.8% (*n* = 128)	1.5% (*n* = 201)	0% (*n* = 2)	0% (*n* = 3)
	Total examined ^1^	*N* = 328	*N* = 391	*N* = 80	*N* = 86

^1^ The total includes individuals that were examined for at least one of the morbidity indicators. ^2^ The global score is the sum of the bladder and the upper tract intermediate scores.

**Table 2 microorganisms-09-01776-t002:** Odds ratio [95% confidence interval] and *p*-values (in parentheses) of hematuria being in category *j* or above (*j* = +, ++ or +++) among children in Senegal in 2016 and 2017, depending on schistosomiasis infection intensity and adjusted for age, sex, and study site. Only individuals with complete data on hematuria, urogenital or intestinal infection intensities, age, sex, and study site were included in the cumulative link logistic regression models (*n* = 509 in 2016; *n* = 472 in 2017).

		Hematuria 2016		
		*Sh* and *Sh-Sb* Urogenital Infection Intensity		
		None	Light	Heavy
*Sm* intestinal infection intensity	None	1	7.19 [5.14–10.1] (*p* < 0.0001)	51.7 [26.4–101] (*p* < 0.0001)
	Light	1.05 [0.46–2.39] (*p* = 0.91)	8.62 [5.09–14.6] (*p* < 0.0001)	70.8 [29.0–173] (*p* < 0.0001)
	Moderate	1.10 [0.21–5.73] (*p* = 0.91)	10.3 [4.30–24.9] (*p* < 0.0001)	97.1 [22.0–429] (*p* < 0.0001)
	Heavy	1.16 [0.10–13.7] (*p* = 0.91)	12.4 [3.51–43.8] (*p* < 0.0001)	133 [15.3–1159] (*p* < 0.0001)
		**Hematuria 2017**		
		*Sh* and *Sh-Sb* urogenital infection intensity		
		None	Light	Heavy
*Sm* intestinal infection intensity	None	1	6.37 [4.33–9.39] (*p* < 0.0001)	40.6 [18.7–88.2] (*p* < 0.0001)
	Light	1.31 [0.53–3.24] (*p* = 0.56)	5.78 [3.37–9.89] (*p* < 0.0001)	25.5 [9.73–66.6] (*p* < 0.0001)
	Moderate	1.72 [0.28–10.5] (*p* = 0.56)	5.24 [2.18–12.6] (*p* = 0.00021)	15.9 [3.13–81.3] (*p* = 0.00086)
	Heavy	2.26 [0.15–34.1] (*p* = 0.56)	4.75 [1.35–16.7] (*p* = 0.015)	9.99 [0.90–111] (*p* = 0.061)

*Sh*: *Schistosoma haematobium*; *Sh-Sb*: *S. haematobium-S. bovis* hybrids; *Sm*: *S. mansoni*. Urogenital infection intensity: none: 0 ep 10 mL (eggs per 10 mL of urine); light: <50 ep 10 mL; heavy: ≥50 ep 10 mL. Intestinal infection intensity: none: 0 epg (eggs per gram of feces); light: 1–99 epg; moderate: 100–399 epg; heavy: ≥400 epg.

**Table 3 microorganisms-09-01776-t003:** Odds ratio (95% confidence interval) and *p*-values (in parentheses) of urinary tract lesions being in category *j* or above as assessed by ultrasound among children in Senegal in 2017, depending on schistosomiasis infection intensity and adjusted for age, sex, and study site. Only individuals with complete data on urinary tract lesions, urogenital or intestinal infection intensities, age, sex, and study site were included in the cumulative link logistic regression models (*n* = 419; except for bladder intermediate score: *n* = 424).

		Global Score (*j* = Positive) ^1^		
		*Sh* and *Sh-Sb* Urogenital Infection Intensity		
		None	Light	Heavy
*Sm* intestinal infection intensity	None	1	6.25 [3.76–10.4] (*p* < 0.0001)	39.0 [14.1–108] (*p* < 0.0001)
	Light	0.57 [0.10–3.22] (*p* = 0.52)	5.96 [2.87–12.4] (*p* < 0.0001)	62.8 [9.18–430] (*p* < 0.0001)
	Moderate	0.32 [0.01–10.4] (*p* = 0.52)	5.69 [1.68–19.3] (*p* = 0.0053)	101 [2.73–3,743] (*p* = 0.012)
	Heavy	0.18 [0.001–33.3] (*p* = 0.52)	5.44 [0.92–31.9] (*p* = 0.061)	163 [0.74–35,730] (*p* = 0.064)
		**Bladder Intermediate Score (*j* = likely or very likely)**		
		*Sh* and *Sh-Sb* urogenital infection intensity		
		None	Light	Heavy
*Sm* intestinal infection intensity	None	1	4.61 [3.04–6.98] (*p* < 0.0001)	21.2 [9.23–48.8] (*p* < 0.0001)
	Light	0.50 [0.12–2.08] (*p* = 0.34)	6.37 [3.36–12.1] (*p* < 0.0001)	80.5 [20.3–319] (*p* < 0.0001)
	Moderate	0.25 [0.01–4.32] (*p* = 0.34)	8.81 [3.07–25.3] (*p* < 0.0001)	306 [25.2–3,703] (*p* < 0.0001)
	Heavy	0.13 [0.002–8.99] (*p* = 0.34)	12.2 [2.69–55.2] (*p* = 0.0012)	1160 [28.8–46,760] (*p* = 0.00018)
		**Upper Tract Intermediate Score (*j* = positive)**		
		*Sh* and *Sh-Sb* urogenital infection intensity		
		None	Light	Heavy
*Sm* intestinal infection intensity	None	1	1.70 [1.03–2.80] (*p* = 0.039)	2.88 [1.06–7.85] (*p* = 0.039)
	Light	0.80 [0.17–3.67] (*p* = 0.77)	1.54 [0.69–3.42] (*p* = 0.29)	2.96 [0.82–10.7] (*p* = 0.098)
	Moderate	0.63 [0.03–13.5] (*p* = 0.77)	1.39 [0.36–5.37] (*p* = 0.63)	3.04 [0.34–26.9] (*p* = 0.32)
	Heavy	0.51 [0.005–49.4] (*p* = 0.77)	1.26 [0.18–8.85] (*p* = 0.82)	3.12 [0.13–77.5] (*p* = 0.49)
		**Bladder Wall Calcification (*j* = present)**		
		*Sh* and *Sh-Sb* urogenital infection intensity		
		None	Light	Heavy
*Sm* intestinal infection intensity	None	1	2.47 [1.49–4.09] (*p* = 0.00045)	6.10 [2.22–16.7] (*p* = 0.00045)
	Light	0.22 [0.04–1.21] (*p* = 0.082)	1.77 [0.79–3.99] (*p* = 0.17)	14.2 [3.67–55.1] (*p* = 0.00012)
	Moderate	0.05 [0.002–1.46] (*p* = 0.082)	1.27 [0.33–4.97] (*p* = 0.73)	33.2 [3.11–355] (*p* = 0.0038)
	Heavy	0.01 [0.0001–1.76] (*p* = 0.082)	0.91 [0.13–6.49] (*p* = 0.93)	77.5 [2.31–2,600] (*p* = 0.015)

*Sh*: *Schistosoma haematobium*; *Sh-Sb*: *S. haematobium-S. bovis* hybrids; *Sm*: *S. mansoni.* Please see Table 2 for infection intensity categorizations. ^1^ The global score is the sum of the bladder and the upper tract intermediate scores.

**Table 4 microorganisms-09-01776-t004:** Cure rate and egg reduction rate in children and adults one month after treatment with praziquantel in two study sites in Senegal in 2016 and 2017 (number of individuals examined indicated in parentheses; only individuals with parasitological data both pre- and post-treatment were included).

Year	Praziquantel Efficacy	Children		Adults	
		Barkedji	Richard Toll	Barkedji	Richard Toll
2016	Cure rate	81% (*n* = 42)	-	81% (*n* = 16)	-
	Egg reduction rate	97.8%	-	100%	-
2017	Cure rate	77% (*n* = 71)	79% (*n* = 258)	62% (*n* = 8)	78% (*n* = 9)
	Egg reduction rate	98.2%	99.5%	99.1%	95.1%

**Table 5 microorganisms-09-01776-t005:** Proportion of children and adults whose morbidity levels decreased one month after treatment with praziquantel in two study sites in Senegal in 2016 and 2017 (number of individuals examined indicated in parentheses; only individuals with anemia, hematuria, or lesions observed using ultrasound pre-treatment were included).

Year	Morbidity Indicator	Children		Adults	
		Barkedji	Richard Toll	Barkedji	Richard Toll
2016	Hematuria	77% (*n* = 35)	-	71% (*n* = 14)	-
	Anemia	-	-	-	-
2017	Hematuria	89% (*n* = 44)	84% (*n* = 195)	86% (*n* = 7)	50% (*n* = 4)
	Anemia	-	51% (*n* = 106)	-	-
	Global score ^1^	63% (*n* = 19)	69% (*n* = 166)	100% (*n* = 1)	75% (*n* = 4)
	Bladder intermediate score	45% (*n* = 22)	66% (*n* = 155)	100% (*n* = 1)	100% (*n* = 3)
	Upper tract intermediate score	100% (*n* = 5)	87% (*n* = 39)	0% (*n* = 1)	0% (*n* = 4)
	Bladder wall calcification	100% (*n* = 6)	59% (*n* = 68)	-	100% (*n* = 2)
	Hepatomegaly	-	-	-	-

^1^ The global score is the sum of the bladder and the upper tract intermediate scores.

**Table 6 microorganisms-09-01776-t006:** Odds ratios (95% confidence intervals) and *p*-values (in parentheses) of decreased levels of morbidity in response to treatment with praziquantel among children in Senegal in 2017, depending on schistosomiasis infection intensity before treatment and adjusted for age, sex, and study site.

Morbidity Indicator	*n*	Type of Infection			
Hematuria 2017	167	*Sh* and *Sh-Sb* urogenital infection intensity			
		Light		Heavy	
		0.76 [0.31–2.27] (*p* = 0.72)		0.57 [0.09–5.17] (*p* = 0.72)	
		*Sm* intestinal infection intensity			
		Light	Moderate		Heavy
		1.29 [0.56–3.28] (*p* = 0.51)	1.67 [0.31–10.8] (*p* = 0.51)		2.17 [0.17–35.4] (*p* = 0.51)
Anemia 2017	78	*Sh* and *Sh-Sb* urogenital infection intensity			
		Light		Heavy	
		1.50 [0.45–3.15] (*p* = 0.73)		2.25 [0.20–9.92] (*p* = 0.73)	
		*Sm* intestinal infection intensity			
		Light	Moderate		Heavy
		1.44 [0.76–2.98] (*p* = 0.24)	2.09 [0.58–8.87] (*p* = 0.24)		3.01 [0.44–26.4] (*p* = 0.24)
Global score ^1^	125	*Sh* and *Sh-Sb* urogenital infection intensity			
		Light		Heavy	
		3.82 [1.32–11.1] (*p* = 0.014)		14.6 [1.74–122] (*p* = 0.014)	
		*Sm* intestinal infection intensity			
		Light	Moderate		Heavy
		1.72 [0.76–3.89] (*p* = 0.19)	2.97 [0.58–15.1] (*p* = 0.19)		5.13 [0.45–58.9] (*p* = 0.19)
Bladder intermediate score	119	*Sh* and *Sh-Sb* urogenital infection intensity			
		Light		Heavy	
		3.80 [1.31–11.0] (*p* = 0.014)		14.5 [1.72–121] (*p* = 0.014)	
		*Sm* intestinal infection intensity			
		Light	Moderate		Heavy
		1.57 [0.73–3.40] (*p* = 0.25)	2.46 [0.53–11.5] (*p* = 0.25)		3.87 [0.38–39.1] (*p* = 0.25)
Bladder wall calcification	46	*Sh* and *Sh-Sb* urogenital infection intensity			
		Light		Heavy	
		1.38 [0.44–4.75] (*p* = 0.59)		1.91 [0.19–22.6] (*p* = 0.59)	
		*Sm* intestinal infection intensity			
		Light	Moderate		Heavy
		1.29 [0.53–4.14] (*p* = 0.60)	1.67 [0.29–17.2] (*p* = 0.60)		2.16 [0.15–71.] (*p* = 0.60)

*Sh*: *Schistosoma haematobium*; *Sh-Sb*: *S. haematobium-S. bovis* hybrids; *Sm*: *S. mansoni*. ^1^ The global score is the sum of the bladder and the upper tract intermediate scores.

## Data Availability

The data presented in this study are available on request from the corresponding authors.

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
