# Peer review of "Hybridized Zoonotic Schistosoma Infections Result in Hybridized Morbidity Profiles: A Clinical Morbidity Study amongst Co-Infected Human Populations of Senegal"

_microorganisms, 2021, doi:10.3390/microorganisms9081776_

Round 1
Reviewer 1 Report
As hybridization amongst parasites is increasing, studies over their impact on morbidity and on the treatment outcome will become more actual. In this direction, the present study paves the way to this situation of concern for public health. The authors made a very well designed and elaborate study, the few limitations being minor.
Reviewer 2 Report
In the manuscript “Hybridized zoonotic Schistosoma infections result in hybridized morbidity profiles: a clinical morbidity study amongst co-infected human populations of Senegal” by Fall et al., aims to assess the impact of infections with hybridized Schistosoma haematobium x S. bovis parasites on morbidity. The study presented covers an important topic as in light of increased occurrence of parasite hybrids and the possibly altered virulence, their impact on human health remains largely unknown. While the article is well written, I still have a few concerns regarding the conclusions for the Sh x Sb hybrids.
The authors used cox1 and ITS profiling of miracidia. As cox1 is a mitochondrial gene and reflects maternal origin, there could be a bias towards detecting the female adult Sh-Sb-hybrids. With respect to the location and the mentioned possibility that dominant male Sm worms could “lure” (hybrid) females in the mesenteric veins, which would explain the increased hepatomegaly in children. What is the frequency of hybrids identified from either ITS or cox1? In general, it would be interesting to know where the adult hybrid schistosomes reside.
Tables 1 and 4 seem to be cut off due to the typesetting.
It is unclear, how the different n-numbers in brackets of the tables are calculated. For example in Table 1, the individual values for children, 2017, BK, range from 99 to 374, but a total number of 396 is indicated. And on what basis were, for example, the 99 children examined for hepatomegaly selected, but all others were not – which also could introduce bias in the presence or absence of morbidity. And why were only 42 included in Figure 2?
Did the authors exclude menstrual blood as the confounder of the increased OR of hematuria in adult females?
The different OR of hybrids vs non-hybrids is rather limited – still important to notice, though. Is there a similar morbidity indicator for adults? In the conclusion, the authors mention exacerbated morbidity. I think this is misleading as it suggests an increased health risk from hybrid infections, but it is only in a small group of examined children, where a significant difference exists.
In general, I would think readers would benefit from additional graphic representation of the results as the extensive data sets in tables is more difficult to grasp. For example, also a figure of how many hybrid infections were detected in all examined persons and among infected persons would be very informative.
For Table 5 – is there a difference between Sh and Sh-Sb?
It is not clear to me, what the benefit of the global score in the tables is – especially for the odds ratios it is unclear how this is calculated as a sum of two scores – wouldn’t that overestimate the differences?
In most tables the “u” of urogenital is formatted in italics. Overall, the formatting of the tables should be carefully checked for bold and italic, and lines.
line 102: reference is missing
